# Optoelectronic Enhancement of Perovskite Solar Cells through the Incorporation of Plasmonic Particles

**DOI:** 10.3390/mi13070999

**Published:** 2022-06-25

**Authors:** Mohamed Salleh Mohamed Saheed, Norani Muti Mohamed, Balbir Singh Mahinder Singh, Mohamed Shuaib Mohamed Saheed, Rajan Jose

**Affiliations:** 1Centre of Innovative Nanostructures and Nanodevices (COINN), Universiti Teknologi PETRONAS, Seri Iskandar 32610, Malaysia; mohamed.salleh87@gmail.com (M.S.M.S.); noranimuti_mohamed@utp.edu.my (N.M.M.); balbir@utp.edu.my (B.S.M.S.); 2Department of Fundamental and Applied Sciences, Universiti Teknologi PETRONAS, Seri Iskandar 32610, Malaysia; 3Department of Mechanical Engineering, Universiti Teknologi PETRONAS, Seri Iskandar 32610, Malaysia; 4Center for Intelligent Materials, Universiti Malaysia Pahang, Kuantan 26300, Malaysia; 5Faculty of Industrial Sciences & Technology, Universiti Malaysia Pahang, Kuantan 26300, Malaysia

**Keywords:** plasmonic perovskite solar cell, plasmonic effects, recombination rate, charge transfer

## Abstract

The optoelectronic advantages of anchoring plasmonic silver and copper particles and non-plasmonic titanium particles onto zinc oxide (ZnO) nanoflower (NF) scaffolds for the fabrication of perovskite solar cells (PSCs) are addressed in this article. The metallic particles were sputter-deposited as a function of sputtering time to vary their size on solution-grown ZnO NFs on which methylammonium lead iodide perovskite was crystallized in a controlled environment. Optical absorption measurements showed impressive improvements in the light-harvesting efficiency (LHE) of the devices using silver nanoparticles and some concentrations of copper, whereas the LHE was relatively lower in devices used titanium than in a control device without any metallic particles. Fully functional PSCs were fabricated using the plasmonic and non-plasmonic metallic film-decorated ZnO NFs. Several fold enhancements in photoconversion efficiency were achieved in the silver-containing devices compared with the control device, which was accompanied by an increase in the photocurrent density, photovoltage, and fill factor. To understand the plasmonic effects in the photoanode, the LHE, photo-current density, photovoltage, photoluminescence, incident photon-to-current conversion efficiency, and electrochemical impedance properties were thoroughly investigated. This research showcases the efficacy of the addition of plasmonic particles onto photo anodes, which leads to improved light scattering, better charge separation, and reduced electron–hole recombination rate.

## 1. Introduction

Solar energy is the most abundant renewable energy available on planet Earth and can be readily harvested by converting it into electrical or thermal energies. A plethora of routes has been explored to harvest clean energy from this vast reservoir as electric power through various generations of solar cells, such as monocrystalline and polycrystalline silicon solar cells, thin-film solar cells, organic solar cells (such as bulk heterojunction cells fabricated from semiconducting polymers), and dye-sensitized solar cells [1,2]. The latest addition in this series is solar cells that use an organic–inorganic hybrid perovskite as the light absorber, known as perovskite solar cells (PSCs) [3,4,5,6]. The emergence of PSCs has been phenomenal, starting from the work of Kojima et al. with a photoconversion efficiency (PCE) of just ~3.8% [7] and skyrocketing at an unprecedented pace to ~23% in a decade [8,9,10,11]. This remarkable achievement was made possible through numerous optimizations from the selection of cations in the photoactive layer and thorough screening of n-type electron-transporting material (ETM) and p-type hole-transporting material (HTM) [12,13,14,15]. A significant number of studies have been carried out in order to simplify PSCs through the removal of HTM [16,17,18] and the refinement of the ETL [3,19,20,21]. As indicated by previous research, PSCs without HTM with a PCE of ~19.3% are reported in diverse configurations in planar and mesoporous architectures [22,23,24,25]. This fresh approach confers the dual benefit of significantly lowering PSC fabrication costs and simplifying the deposition processes. In addition, Chen et al. further advanced perovskite crystallinity through the use of modified deposition techniques capable of producing PSCs without HTM with high efficiency [26]. Furthermore, materials merged with halide precursors through an optimized precursor deposition technique are capable of further augmenting the PCE via prolonging the carrier diffusion length and improving electron–hole transportation within the perovskite crystals [27,28,29,30,31,32,33,34,35,36,37].

Various alternative strategies have also been attempted to maximize the light-harvesting capability of PSCs, such as optimization and modification of the ETL [38,39,40]. One valid approach is the incorporation of plasmonic metals (PMs) into the ETL, which promotes light concentration and leads to better light scattering and electron–hole disassociation [41,42,43,44]. PMs such as gold (Au), silver (Ag), and copper (Cu) give rise to vivid colors due to their localized surface plasmon resonance (LSPR) [42,45,46,47,48]. The LSPR lends merit to the solar cells via light scattering and the absorbing layers [42,46,47,48,49,50,51,52,53]. The resulting substantial increase in light scattering can be attributed to the non-radiative decay of the PMs transferred from the light absorption layer to the main hot electron–hole pairs, which are then immediately promoted to the secondary electron–hole pair [54,55,56,57,58,59]. This effective electron–hole pair generation can be further exploited through the integration of these plasmonic particles into the ETL in photo-catalytic and light-harvesting devices besides the tandem formation of Schottky’s barrier between metal oxides and the PMs [60,61,62,63,64,65,66]. The barricade acts as an efficient charge separation barrier at the conduction band of the metal oxides that selectively restricts recombination between the electron and the hole. One of the critical issues in PSCs is their inferior operational stability; however, recent studies have shown that the plasmonic local heating effect could enhance the stability of the device [67]. In addition, PMs are reported to be excellent light-trapping devices when coupled with solar cells [68], light controlling in metal–dielectric–metal waveguides [69], design of tunable multi-band perfect metamaterial absorbers [70], highly sensitive refractive-index sensors based on strong magnetic resonance in metamaterials [71], etc., thereby offering numerous futuristic applications of the solar cells employing them.

In this study, metal nanoparticle (Ag, Cu, and Ti)/metal oxide nanostructure (ZnO) interfaces were fabricated, and the effect of plasmonic (Ag and Cu) and non-plasmonic (Ti) metal particles on the photocurrent density (*J_SC_*) and open-circuit voltage (*V_OC_*) of PSCs was examined. Herein, the metal/ZnO nanoflower (NF) structures were developed with different deposition parameters to serve as the ETL in the plasmonic perovskite solar cells (PPSCs). The surface morphologies of PPSCs were thoroughly investigated to comprehend the optoelectronic properties of the fabricated devices by examining *J_SC_*, *V_OC_*, light-harvesting efficiency (LHE), and steady-state photoluminescence (PL). The correlation between PM nanoparticle concentration and ZnO NFs was determined by evaluating the incident photon-to-current conversion efficiency (IPCE), wherein the addition of PMs enhanced electron–hole generation and improved the charge disassociations. Considerable increases in *J_SC_* and *V_OC_* were also observed in Ag- and Cu-deposited PPSCs. The optimized PPSCs with Ag and Cu delivered a maximum PCE of ~8.3% and ~6.41%, respectively, compared with pristine PSCs, which had a mere ~2.41% PCE. Therefore, the enhancement of PCE can be attributed to the improvement in *J_SC_*, *V_OC_*, LHE, and the charge separation mechanism within the PM/ZnO NF interfaces. This study opens a new avenue for enhancing the optoelectronic properties of the electron transporting layer for artificial photosynthetic and photocatalytic devices.

## 2. Experimental Details

First, fluorine-doped tin oxide (FTO) glasses were laser-scribed into a square configuration and cleaned thoroughly through ultra-sonication in deionized (DI) water, acetone, and ethanol disparately in sequence. The electron-transporting material, ZnO, was prepared as reported in a previous study [72]. In a typical experiment, ~2.195 g of zinc acetate dehydrate [Zn (CH_3_COO)_2_.2H_2_O] was dissolved in 50 mL of ethanol and stirred vigorously at 60 °C for 30 min to obtain a sol. Next, monoethanolamine was constantly added dropwise into the mixture at 10 min intervals for a period of 2 h to stabilize the solution. The synthesized ZnO solution was then stored overnight in a dehumidified environment at room temperature prior to seed layer deposition. A compact ZnO thin film was prepared by spin-coating the above sol on cleaned FTO glasses at 3000 rpm for 30 s. The entire procedure was repeated three times to obtain a thick and consistent film. The ZnO-coated FTO glass was then sintered at 300 °C for 1 h and subjected to hydrothermal processing to prepare ZnO nanoflowers (ZnO NFs). In tandem with the sintering process, a growth solution was synthesized by reacting zinc nitrate hexahydrate (Zn (NO_3_)_2_·6H_2_O) with hexamethylenetetramine in DI water. The ZnO thin films prepared earlier were then immersed in the growth solution with the reactant surface of the ZnO thin films facing the base of the reaction chamber. Then, both the growth solution and treated ZnO thin film were transferred into a vacuum oven in a Teflon chamber for the hydrothermal process at 95 °C for 4 h. The synthesized ZnO NFs were then rinsed with DI water and sintered at 300 °C for 1 h.

Next, plasmonic (Ag and Cu) metals and non-plasmonic (Ti) metals were deposited on ZnO NFs through the direct current (DC) sputtering method employing corresponding metal targets: Ag (99.99% purity), Cu (99.99% purity), and Ti (99.99% purity). Initially, the pressure of the sputtering chamber was reduced to a base pressure of 10^−7^ Pa, and later, the working pressure was increased and maintained at 10^−3^ Pa by regulating the pure Argon pressure. The metal nanoparticles of varying sizes were deposited at a power of ~100 W by varying the sputtering times of 5 s, 7 s, and 10 s. All the sputtered ZnO NF samples were kept in separate vacuum antechambers at 10^−3^ Pa prior to use to prevent the oxidation of the materials being studied.

The perovskite thin film was deposited by adopting a two-step deposition process. First, ~462 mg of PbI_2_ powder was diluted in 1 mL of DMF to prepare a standard solution of 1 M PbI_2_. The standard solution was then continuously stirred at 70 °C for 2 h. In tandem with the preparation of the PbI_2_ solution, methylammonium iodide solution was prepared by diluting 100 mg of the powder in 10 mL of 2-propanol. The PbI_2_ solution was then spin-coated on a preheated FTO glass coated with ZnO NFs at 2000 rpm for 30 s and later annealed at 70 °C for 30 min. The samples were left to cool down to room temperature and then immersed in CH_3_NH_3_I solution for 20 s to form crystalline perovskite. The synthesized crystals were annealed at 70 °C for 30 min and left to cool down to room temperature. Subsequently, 3D graphene was fabricated as described in our previous work [41] and utilized as the cathode in this study rather than the commonly used Au. The 3D graphene electrode was thermally pressed and mechanically clamped with the device. All device fabrications were performed in a controlled environment in which the relative humidity was set at <1%, while O_2_ was set well below 200 ppm to obtain highly crystalline perovskites.

The chemical composition and surface morphology of the synthesized samples were investigated via field emission scanning electron microscopy (FESEM) using a Zeiss Supra55 VP (Oberkochen, Germany). The surface chemical structure of ZnO NFs was studied by X-ray photoelectron spectroscopy (XPS) with a K-Alpha system, Thermo Scientific (Waltham, MA, USA), utilizing Al-K alpha radiation with a spot size of 400 µm. The phase and crystallinity of the materials were assessed through X-ray diffraction (XRD) recorded on an X’Pert MPDPRO, PANalytical (Cu anode, 1.5406 Å) (Malvern, UK). A step size of 0.01 deg and an acquisition time of 1s/deg was used for recording the XRD pattern. Optical transmittance and absorbance (UV–Vis) were measured using a Perkin-Elmer (Waltham, MA, USA) Lambda 950 spectrophotometer with a slit width of 2 nm at normal incidence, while the photoluminescence of the samples was evaluated using a Horiba Fluorolog-3, HORIBA Jobin Yvon (Kyoto, Japan), with the excitation level set at 600 nm.

The current density–voltage (*J–V*) curves of the devices were measured by subjecting the devices to AM 1.5 G 1 Sun (100 mW/cm^2^) irradiation simulated by a Dyesol illumination lamp (450 W Xenon light source) and measured using a digital source meter (Keithley 2420, Cleveland, OH, USA). The efficiency of the solar cells was further obtained by masking an area of 0.09 cm^2^. The IPCE spectra were measured using an IPCE system, and all data were collected in DC mode with the monochromatic light source ranging from a wavelength of 200 nm to 800 nm. The electrochemical impedance spectroscopy (EIS) data were obtained through the utilization of a Gamry potentiostat in a darkened environment with the applied DC voltage ranging from 0 to1 V and 20 mV AC put on the DC bias voltage at a frequency ranging from 0.1 to 300 k Hz. The *J–V*, IPCE, and EIS characterizations were consistently performed in an ambient environment with a humidity of <70%.

## 3. Results and Discussion

The structural characterization of the ZnO nanorods grown on FTO was given in our earlier publications [41,43]. Herein, ZnO was adorned with metals possessing different intrinsic properties and, in this study, Ag and Cu were selected to represent PMs, whereas Ti was utilized as the non-PM. The materials were meticulously prepared by varying the sputtering duration of the metals in short bursts of 5 s, 7 s, and 10 s whilst maintaining uniformity. Henceforth, the abbreviation ZnO-Ag (5, 7, and 10), ZnO-Cu (5, 7, and 10), and ZnO-Ti (5, 7, and 10) are used throughout the text to represent corresponding samples. The surface morphologies and thicknesses of the decorated ZnO nanostructures are shown in detail in Figure 1. The ZnO nanostructures grown on FTO glass exhibited a flower-like conformation consisting of nanorod arrays, visible in the SEM images shown in Figure 1a–c. These ZnO nanostructures are referred to as ZnO NFs in reference to their floral counterparts. As seen in the images, the average length of the synthesized ZnO NFs was 410 nm within a growth period of 4 h. Appendix A, on the other hand, shows the EDX spectrums of ZnO NFs deposited with Ag, Cu, and Ti, which showcase trace elements of zinc (Zn), oxygen (O_2_), silver (Ag), copper (Cu), and titanium (Ti). The clear peaks of Ag, Cu, and Ti as well as Zn and O peaks validate the presence of plasmonic and non-plasmonic nanoparticles on the surface of the ZnO NFs. The absence of impurities in the fabricated samples indicates the phase purity of the ZnO NFs. Furthermore, the cross-section image (Figure 1d) depicts in detail the formation of a compact and dense ZnO seed layer, which played a pivotal role in preventing direct contact between the FTO and the cathode terminal of the device.

To characterize the elemental composition of the deposited metals on ZnO NFs and their corresponding binding energies, XPS analyses were performed. Figure 2a displays the XPS spectra of ZnO-Ag, ZnO-Cu, and ZnO-Ti, in which the full scan unveiled the presence of the elements Zn, Ag, Cu, Ti, and O_2_. The Zn main peak, Zn2p, exhibited binding energies at 1022.6 eV and 1045.8 eV, evincing the presence of Zn^2+^ [73]. Figure 2b displays Ag metal binding energies at 368.48 eV and 374.48 eV [48], whereas Cu displayed binding energies at 932.48 eV and 952.28 eV, as shown in Figure 2c [74]. Ti had binding energies at 459.48 eV and 464.98 eV [75], as shown in Figure 2d. Interestingly, when the small chemical shifts in binding energy in Ag, Cu, and Ti NPs were compared with the binding energies of bulk Ag (368.2 eV), Cu (933 eV), and Ti (458.5 eV), a substantial electronic bond polarization between NPs and ZnO NFs could be observed.

Optical characteristics, such as the reflectance and absorbance of the fabricated devices, are plotted in Figure 3, which includes the spectrum of the pristine ZnO NFs in addition to the sputtered devices ZnO-Ag (5, 7, and 10), ZnO-Cu (5, 7, and 10), and ZnO-Ti (5, 7, and 10). The ZnO NFs exhibited a maximum reflectance of 55% in the visible range, with wavelengths between 380 nm to 690 nm, whereas ZnO-Cu (5, 7, and 10) and ZnO-Ti (5, 7, and 10) indicated an ameliorated reflectance of up to 25% higher compared with ZnO NFs. On the contrary, ZnO-Ag (7 and 10) possessed a lower reflectance percentage than ZnO NFs. An exception to this, however, was ZnO-Ag5, which showed superior reflectance in the redshift region from 500 nm to 690 nm. It is important to note that a stable reflectance plateau could be observed in the visible range for ZnO-Cu (5, 7, and 10) and ZnO-Ti (5, 7, and 10) thin films. The absorbances of ZnO NFs, ZnO-Ag (5, 7, and 10), ZnO-Cu (5, 7, and 10), and ZnO-Ti (5, 7, and 10) are charted in Figure 3b. Strong band edge absorbance at 373 nm of hexagonal ZnO NFs is visible in Figure 3b.

The band gap energies of the ZnO NFs, ZnO-Ag (5, 7, and 10), ZnO-Cu (5, 7, and 10), and ZnO-Ti (5, 7, and 10) were calculated using the Kubelka–Munk method and by plotting a Tauc graph. The absorbance coefficient (*a*) was derived by using the formula $a=\frac{A{\left(h\upsilon -E_{g}\right)}^{1/2}}{h\upsilon }$, where *E_g_* is the energy band gap, *A* is the constant, *v* is the incident radiation frequency, and *h* is Planck’s constant. Figure 4 divulges the bandgap energies *hv* versus (*ahv*)^2^, which provides the absorbance coefficient. Peculiarly, the band gap perceived for both ZnO NFs and ZnO-Ag7 was ~3.2 eV, whereas both ZnO-Cu7 and ZnO-Ti7 had a band gap at ~3.25 eV. The band gap energies of ZnO NFs, which were within the UV wavelength region, implied the presence of high photo-catalytic activity. No significant deviation in the band gaps was visible for the Ag-, Cu-, and Ti-deposited ZnO NFs compared with the pristine ZnO NFs, which exhibited approximately similar band gaps to the TiO_2_ nanorods.

Functional control for both the plasmonic and non-plasmonic devices was conferred to them through the deposition of the perovskite light absorber via a two-step sequential deposition method. Figure 5 depicts the surface morphologies and the cross-section images of the perovskite layer on ZnO NFs, which functioned as the ETL. The apex view revealed homogenously deposited perovskites that covered the surface of ZnO NFs entirely. The elements within the ZnO NF structures were validated by utilizing energy dispersive X-ray inspection, as shown in Appendix A. The cross-sections of ZnO NFs with perovskite demonstrated outstanding pore infiltration in the ZnO nanostructure, and its role as a capping layer enabled the maximization of LHE.

As elucidated earlier, the PPSC structure consisted of ZnO NFs decorated with PMs capable of improving *J_SC_*, *V_OC_*, and *FF*. These enhancements were intended by-products of Schottky’s barrier, which diminished the electron–hole recombination rate and thus led to better electron–hole separation in addition to increased induction of electrons owing to the influence of the PM NPs. To further analyze the impact of PMs on semiconductor-based perovskite solar cells, Ag, Cu, and Ti decorated ZnO NFs were fabricated to function as n-type semiconductors (ETL).

The optical properties of the perovskite/ZnO NF composite depicted in Figure 6 were scrutinized by studying the PL properties, reflective capability, and absorptive capacity. The reflective and absorptive capabilities of ZnO NFs coated with Ag, Cu, and Ti were plotted in Figure 6a,b, respectively. Figure 6a shows that ZnO-Ag exhibited relatively high reflectance compared with the ZnO-Cu and ZnO-Ti sets. Delving further, ZnO-Cu showed superior reflectance compared with ZnO-Ti; the latter showed reflectance well below that of the unmodified ZnO device. These phenomena comply with the Drude model, in which the optical properties of metal nanoparticles rise from free electron scattering, which gives high reflectance and low transparency. The Ag NPs integrated into the ZnO NF matrix work through the confinement and reflection of the light back into the light absorber, which elevates the absorbance index to its highest point, and this index is closely trailed by Cu- and Ti-integrated ZnO NFs. The light absorption capabilities of the perovskite/ZnO NF composite are vital in discerning the intrinsic quality of LHE in both plasmonic and non-plasmonic nanostructures. The ZnO-Ag group demonstrated high absorptive capacity compared with the pure ZnO NF, ZnO-Cu, and ZnO-Ti groups. All the synthesized ZnO (Ag, Cu, and Ti) nanostructures exhibited absorbance at 760 nm, which coincides with the well-established perovskite layer absorption wavelength.

The LHE of the samples was appraised and charted in Figure 6c utilizing the already discussed reflectance and absorbance data. The data revealed that the entirety of the ZnO-Ag group and ZnO-Cu (5 and 7) possessed higher LHE compared with pure ZnO NFs. Conversely, ZnO-Cu10 and the ZnO-Ti group altogether exhibited markedly lower LHE than pure ZnO NFs. The glaring advantage of ZnO-Ag emerges from the intrinsic optical properties of the plasmonic Ag NPs, which have a relatively high reflective and absorption index compared with the Cu and Ti nanostructures. Figure 6d illustrates the effects of excited electron transfer and electron–hole recombination of Ag, Cu, and Ti integrated with ZnO along with perovskite as light absorbers, which were studied using steady-state PL. It was also evident that PL quenching at the interfaces of the perovskite/ZnO-Ag5, ZnO-Cu, and ZnO-Ti groups as well below that of the perovskite/ZnO device, which indicates excellent charge transfer.

The electrical characteristics of the ZnO-(Ag, Cu, and Ti) PSCs were examined via current density–voltage measurement, IPCE, and EIS. The entire set of *J–V* parameters, namely *V_OC_*, *J_SC_*, fill factor (FF), and PCE, are summarized in Table 1. Major *V_OC_* and *J_SC_* enhancements were achieved in ZnO-Ag5 compared with pristine ZnO NFs, with values as high as ~17.58 mA/cm^2^ and ~0.99 V, respectively. Conversely, ZnO-Ag7 and ZnO-Ag10 photoanodes displayed lower FF compared with pristine ZnO NFs. This can be attributed to the -PL quenching, in which ZnO-Ag10 had similar quenching capabilities to pristine ZnO, although it is important to note that ZnO-Ag7 had slightly lower quenching than pristine ZnO. Clearly, the superior performance of ZnO-Ag5 was due to its relatively substantial PL quenching in comparison with ZnO NFs, ZnO-Ag7, and ZnO-Ag10. Although the photoanode constructed from ZnO-Cu and ZnO-Ti had a great advantage in the form of high *V_OC_*, the photoanode underwent drastic *J_SC_* suffering. Primarily, *J_SC_* is dependent on the rate of light excited electron generation and diffusion length, whereas *V_OC_* relies on the rate of electron–hole recombination in solar devices, and as such, ZnO-Cu and ZnO-Ti groups exhibit high *V_OC_*. The increase in the concentration of Ag NPs in ZnO-Ag7 and ZnO-Ag10 leads to the promotion of the electron–hole recombination rate in spontaneous response to the multiplication of electron–hole recombination sites within the ZnO photoanodes, which inadvertently devastates *J_SC_* and *V_OC_*. The corresponding IPCE curves in Figure 7b exhibit the derived *J_SC_* data from *J–V* measurements and showcase the products of LHE, electron generation, and collection efficiency of the solar device in the visible light range. The IPCE for all the solar devices rose steadily from 300 nm, became stabilized in the 400 nm to 700 nm range, and plummeted at the 800 nm mark. ZnO-Ag5 boasted the highest IPCE with a reading of ~41% and was trailed immediately by ZnO-Cu5 at ~39%, while pristine ZnO NFs took the middle ground with an IPCE of ~40%. Strikingly, the ZnO-Ti group showed a lower IPCE compared with the ZnO-Ag and ZnO-Cu pair by virtue of the non-plasmonic effects of Ti NPs. Owing to the increase in the LHE of the Ag analogs, corresponding enhancements in the PCE could be observed. The PSC fabricated using the ZnO-Ag 5 ETL showed clear enhancements in the internal quantum efficiency at the violet and red ends of the spectrum. This enhancement in IPCE and internal quantum efficiency clearly increased the photocurrent density, as shown in Figure 7a.

Electrochemical impedance was utilized to establish the interfacial series resistance, charge transfer resistance, and variation correlation for the acquired *J_SC_* and *V_OC_*. The Nyquist plot displays two prominent semicircles with one at a high frequency and the other at a much lower frequency. The primary semicircle is visible at a high frequency principally due to the charge transfer resistance between the cathode and the electrolyte. The second semicircle, which occurs at a low frequency, is the evident consequence of charge transfer resistance between FTO/ZnO NFs and perovskite. Perturbingly, the transmission line (TL) behavior ascribed to electron transport resistance was heavily suppressed on account of exceedingly thin ZnO NFs. By way of explanation, immensely low electron transport resistance results in TL non-appearance. Figure 8a–c show the Nyquist plot for the ZnO (Ag, Cu, and Ti) groups, in which the internal resistance of perovskite/ZnO-PMs can be ascertained by the diameter of the impedance semicircle. The semicircle diameter of ZnO-Ag5 is the shortest among the photoanode configurations, which signifies the comparatively low electron–hole recombination rate of the photoanode.

Figure 8d exhibits the schematic circuit consisting of series resistance (R_S_), parallel charge transfer resistance (R_CT_), and constant phase capacitance (CPE). R_S_ draws its resistance from the interface between FTO/ZnO NFs and ZnO NFs/perovskite; meanwhile, R_CT_ and CPE owe their resistance to the perovskite/3D graphene interface. The investigated parameters, namely R_S_, R_CT_, and CPE, are tabulated in Table 2. Deriving from Table 2, the ZnO-Ag group had the closest R_CT_ to the ZnO device, whereas the ZnO-Cu and ZnO-Ti groups demonstrated a sizeable R_CT_. These findings validate the fact that the integration of non-plasmonic NPs into photoanodes boosts charge transfer resistance and, in turn, promotes the electron–hole recombination rate and reduces the electron transfer efficiency and the short-circuiting of current.

Predominantly, light-absorbing materials such as perovskites absorb photon energy, resulting in the generation of electron–hole pairs in which the electrons are extracted by means of the ETL (TiO_2_ and ZnO), while the holes are actively segregated by the cathode. The electron and hole recombine in a continuous cycle at the cathode, with the completion of the circuit using an external load. Figure 9 illustrates the mechanisms of electron–hole generation, separation, and transportation of plasmonic Ag NPs integrated within ZnO NFs. The incorporation of PM NPs onto ZnO NFs entails a condition whereby the Fermi level of the individual PM NPs closely mimics the Fermi level of the semiconductor surface. High light absorbance is perceived in PPSCs owing to the immensely anticipated LSPR effects of PMs that promote induced hot electrons, multiplication of secondary electron–hole generations, near-field enhancement, and non-radiative decay energy transfer. In this study, Ag and Cu NPs on ZnO NFs were employed to enhance the light absorption properties of the solar device through the precise manipulation of LSPR effects. Primarily, PM (Ag and Cu)-generated surface plasmons (hot electrons) intensified electron densities and amplified the main hot electrons (from perovskites) to induce secondary electron generation via resonant energy transfer. The secondary electrons were subsequently injected into the ZnO NFs’ conduction band (CB). The electrons generated from the lower unoccupied molecular orbit (LUMO) of perovskite were subjected to a twofold process in which the electrons were transported to the ZnO NFs’ CB and leveled out on the surface of PMs. The additional electrons on the surface of Ag and Cu NPs led to a more balanced Fermi level in both ZnO NFs and PMs. The balanced Fermi level is crucial to stimulate the rapid transportation of electrons from PM NPs into the ZnO NFs’ CB. Additional leverage that surfaced from the integration of PMs with ZnO NFs was the formation of Schottky’s barrier, which significantly reduced electron–hole transport from ZnO NFs CB to the highest occupied molecular orbital (HOMO) in perovskite. The restriction on electron reactions also applied under inverse conditions, in which the electrons were unable to traverse from the surface of the PM to the HOMO level in perovskite or from the CB of the ZnO NFs to the Fermi level. Owing to the above factors, effective charge separation was observed, as evident in the case of the integrated ZnO–Ag5, which showed the highest photocurrent density with ZnO-Cu5 following closely, and these verify the effectiveness of electron transfer from the PMs to the ZnO NFs’ CB. However, an increase in the deposition time of Ag and Cu NPs led to a decrease in the density of the generated photocurrent. This phenomenon can be attributed to the loss of electrons during recombination, in which ZnO–Ag (7 and 10) and ZnO–Cu (7 and 10) displayed reduced *J_SC_*. Furthermore, ZnO NFs integrated with Ti NPs showcased a lower recombination rate, with an increase in the Ti NPs’ deposition time from 5 s to 10 s as an increase in *J_SC_* occurred due to a severe reduction in the generation of electrons, as displayed in Table 1.

Therefore, the integration of PM NPs with ZnO NF solar cells reap a myriad of benefits, as the light-harvesting capability is substantially boosted in the face of LSPR near-field enhancement, induction of surface plasmons, secondary electron generation via resonant energy transfer, and the formation of Schottky’s barrier, which restricts the reversal of the electron transport and, in turn, lowers the recombination rate. Despite this, adverse effects, such as a high electron–hole recombination rate and a reduced photocurrent density, were actively witnessed at high PM concentrations.

## 4. Conclusions

An efficient LSPR-based PPSC was successfully developed through the synthesis of PM (Ag and Cu)-decorated ZnO NFs that exhibited excellent optoelectronic performance. The integration of PMs, such as Ag and Cu, into the solar device augmented the light-harvesting efficiency, lowered the charge recombination rate, and improved *J_SC_*. which reveals new avenues for further PSC performance enhancement. The optical enhancements of LHE in ZnO–Ag-integrated perovskite were demonstrably high and stable with a broad wavelength ranging from 750 nm to 400 nm, whereas ZnO–Cu and ZnO–Ti showed stable LHE in the comparatively narrower range of 550 nm to 400 nm. Moreover, the high quenching rate of Ag NPs embedded in ZnO indicated excellent electron transportation capability and an exceptionally low charge transfer resistance, as validated through EIS studies. The integration of PMs into ZnO reduces charge transfer resistance and enhances electron–hole generation, intensifies electron–hole separation, and forms Schottky’s barrier, which significantly reduced the electron–hole recombination rate. This superior performance was most apparent in ZnO–Ag5, which delivered a *J_SC_* of ~17.83 mA.cm^−1^ and a PCE of ~8.3%, which implies that the fabricated ETL is capable of being utilized in various artificial photosynthesis devices.

## Figures and Tables

**Figure 1 micromachines-13-00999-f001:**
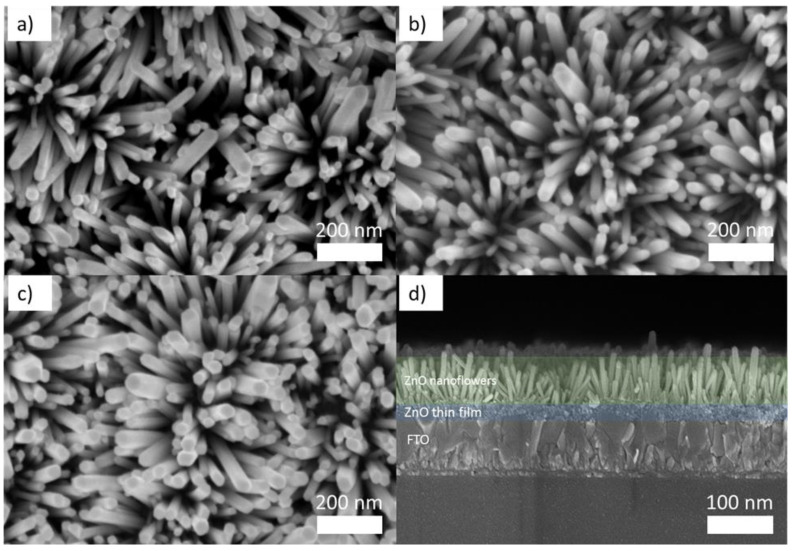
SEM images depicting the top view of ZnO NFs sputtered with (**a**) Ag, (**b**) Cu, and (**c**) Ti as well as (**d**) SEM image of the thin-film cross-section.

**Figure 2 micromachines-13-00999-f002:**
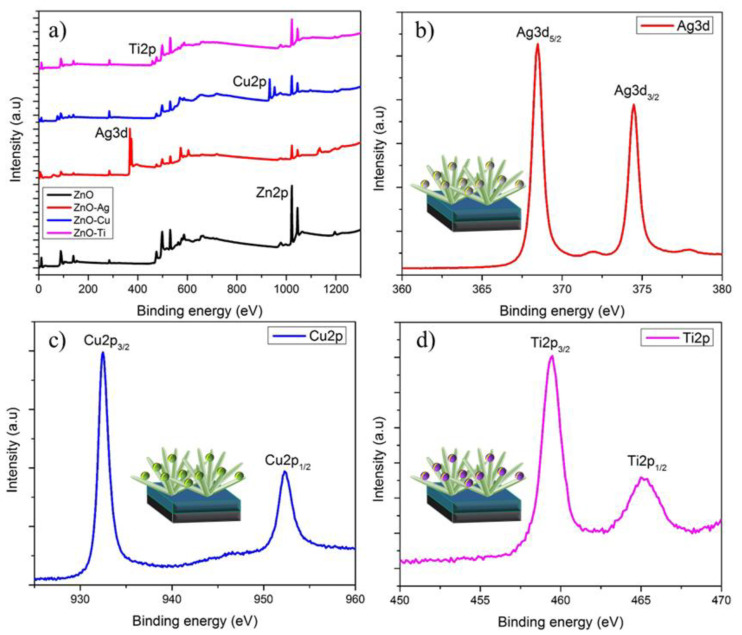
(**a**) XPS survey spectra of ZnO NFs deposited with Ag, Cu, and Ti; (**b**) Ag XPS scan image; (**c**) Cu XPS scan image; (**d**) Ti XPS scan image.

**Figure 3 micromachines-13-00999-f003:**
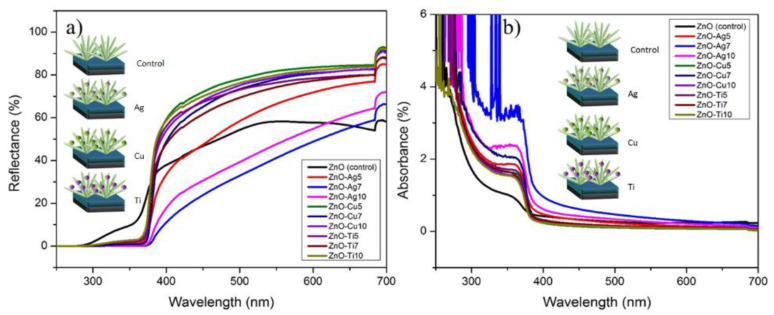
UV-visible spectrum charts of ZnO NFs, ZnO-Ag (5, 7, and 10), ZnO-Cu (5, 7, and 10), and ZnO-Ti (5, 7, and 10) with chart (**a**) displaying reflectance and chart (**b**) displaying absorbance.

**Figure 4 micromachines-13-00999-f004:**
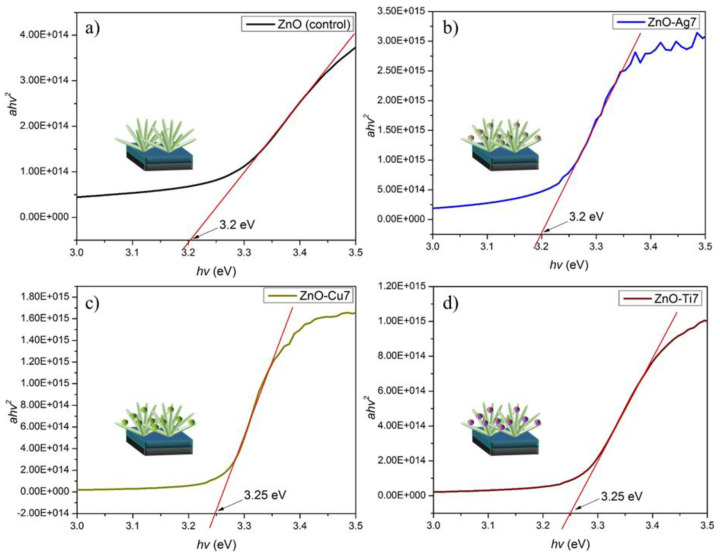
Band gap energies of (**a**) ZnO NFs (control), (**b**) ZnO-Ag7, (**c**) ZnO-Cu7 and (**d**) ZnO-Ti7.

**Figure 5 micromachines-13-00999-f005:**
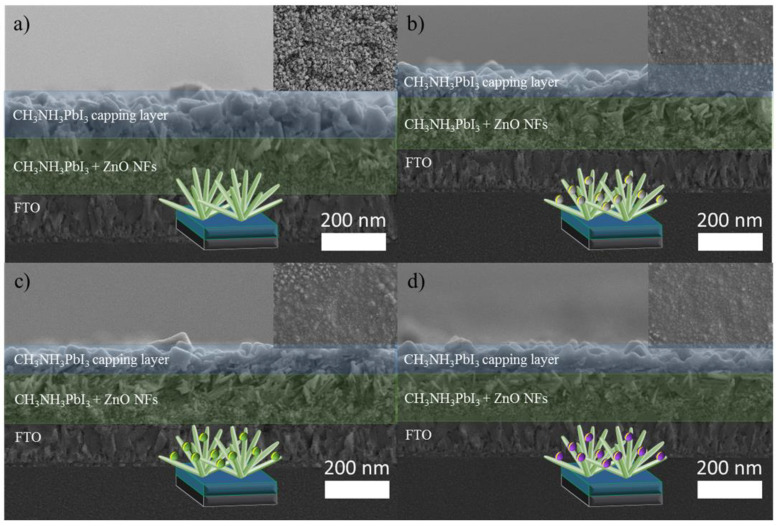
Images of SEM cross-sections and the respective surface morphology inset of CH_3_NH_3_PbI_3_ (perovskite) deposited on (**a**) ZnO NFs, (**b**) ZnO-Ag7, (**c**) ZnO-Cu7, and (**d**) ZnO-Ti7.

**Figure 6 micromachines-13-00999-f006:**
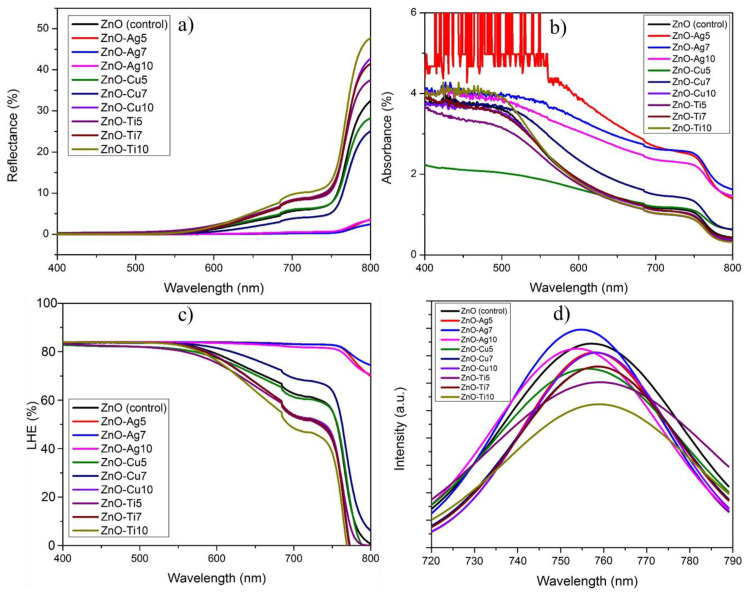
ZnO (Ag, Cu, and Ti) coated with perovskite depicting (**a**) reflectance, (**b**) absorbance, (**c**) LHE, and (**d**) steady-state photoluminescence.

**Figure 7 micromachines-13-00999-f007:**
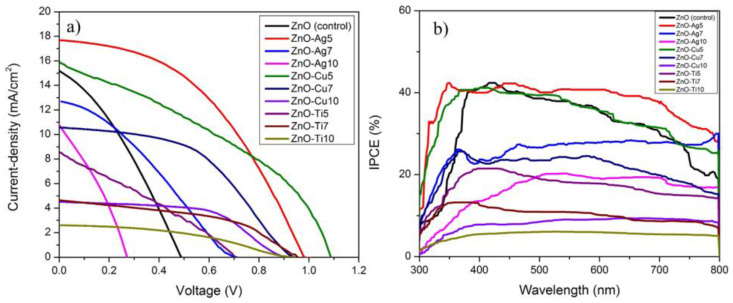
(**a**) *J–V* curves of pristine ZnO NFs and ZnO (Ag, Cu, and Ti) and (**b**) IPCE curves of pristine ZnO NFs and ZnO (Ag, Cu, and Ti) PSCs.

**Figure 8 micromachines-13-00999-f008:**
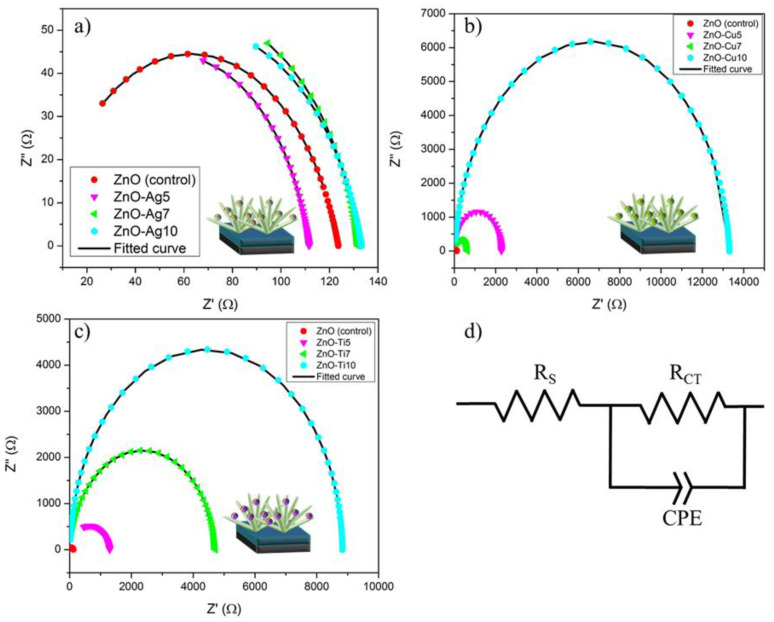
EIS and fitted curves for (**a**) ZnO-Ag (5, 7, and 10), (**b**) ZnO-Cu (5, 7, and 10), (**c**) ZnO-Ti (5, 7, and 10), and (**d**) EIS schematic circuit.

**Figure 9 micromachines-13-00999-f009:**
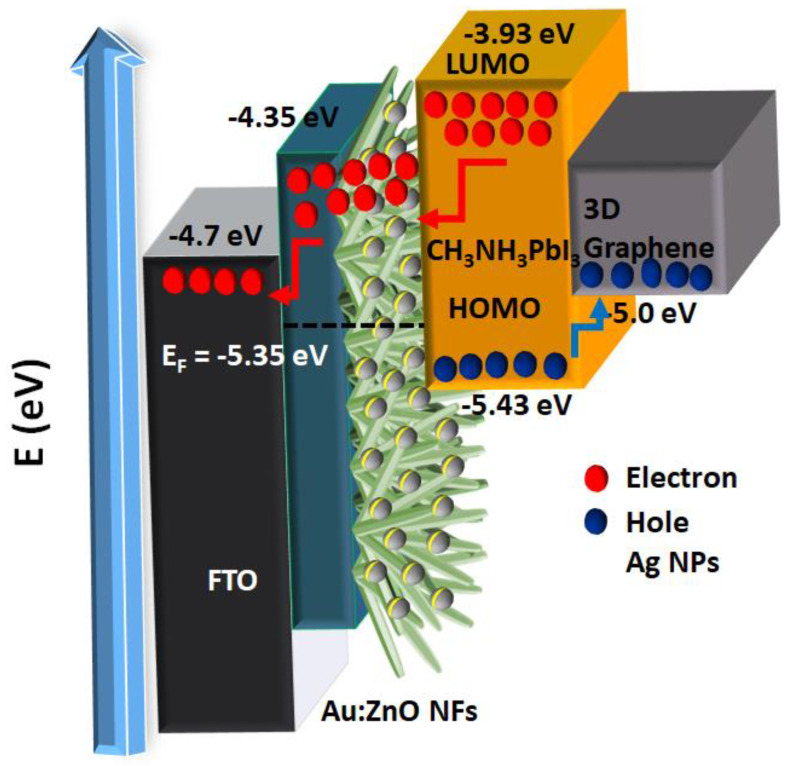
Electron-hole separation and transportation mechanism of plasmonic Ag NPs integrated within ZnO NFs.

**Table 1 micromachines-13-00999-t001:** PSC parameters consisting of current density (*J_SC_*), open voltage (*V_OC_*), fill factor (FF), and power conversion efficiency (PCE).

	*J_SC_* (mA/cm^2^)	*V_OC_* (V)	FF	PCE (%)
ZnO NFs (control)	15.08	0.49	0.33	2.41
ZnO-Ag5	17.83	0.99	0.47	8.30
ZnO-Ag7	12.69	0.71	0.31	2.82
ZnO-Ag10	10.90	0.27	0.31	0.93
ZnO-Cu5	16.00	1.09	0.37	6.41
ZnO-Cu7	10.58	0.96	0.47	4.78
ZnO-Cu10	4.53	0.92	0.54	2.27
ZnO-Ti5	8.60	0.70	0.28	1.75
ZnO-Ti7	4.65	0.96	0.46	2.05
ZnO-Ti10	2.61	0.94	0.40	0.98

**Table 2 micromachines-13-00999-t002:** EIS raw and fitted curve parameters for ZnO (Ag, Cu, and Ti).

	R_S_ (Ω)	R_CT_ (Ω)	CPE (µF)	n
ZnO NFs (control)	2.29 × 10^−11^	121.38	44.0	0.805
ZnO-Ag5	1.58 × 10^−11^	111.61	302	0.85
ZnO-Ag7	7.45 × 10^−12^	131.74	156	0.86
ZnO-Ag10	7.45 × 10^−12^	133.21	326	0.82
ZnO-Cu5	1.57 × 10^−11^	2284.4	2646	0.95
ZnO-Cu7	1.13 × 10^−11^	637.14	1948	0.95
ZnO-Cu10	21.84	13287	3190	0.95
ZnO-Ti5	0.00021	1297.1	4751	0.86
ZnO-Ti7	4.05	4709.5	4462	0.94
ZnO-Ti10	18.98	8811.3	2763	0.99

## Data Availability

All data for this study have been experimentally/theoretically generated and have been included in this paper.

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
