# Peer review of "Optoelectronic Enhancement of Perovskite Solar Cells through the Incorporation of Plasmonic Particles"

_micromachines, 2022, doi:10.3390/mi13070999_

Round 1
Reviewer 1 Report
The manuscript is devoted to studying the possibilities of improving the characteristics of perovskite colar cells (PSС) due to incorporation of plasmonic particles. In this direction, the authors demonstrated impressive results. Moreover, it should be noted the deep and comprehensive study of the details associated with the manufacturing technology of PSC using plasmonic particles, their characterization, as well as understanding the reasons for the differences in the parameters of such devices compared to control non-plasmonic PSCs. Perhaps the only remark may be related to the lack of data on the long-term stability of the characteristics of perovskite solar cells modified by plasmonic particles. It seems that the addition of plasmonic particles can contribute to the gradual degradation of ZnO nanoflowers over time, especially under high light conditions. I would like the authors to comment on this possibility.
Author Response
Many thanks for your time with this manuscript. The issue you have raised is very critical in the case of perovskite solar cells. Considering your point, the manuscript has been revised to include a sentence as the last sentence in the second paragraph of the introduction, which reads as:
"One of the critical issues in PSCs has been their inferior operational stability; however, recent studies have shown that plasmonic local heating effect could enhance the stability of the device [67]."
Reviewer 2 Report
The authors presented a ZnO nanoflowers for the perovskite solar cell. The submitted data and discussion is solid enough and impressive. The novelty maybe is trivial. However, the proposed study is quite comprehensive.
Based on the reviewer's opinion, this manuscript is publishable in micromachine
Author Response
Many thanks for your time with this manuscript!
Reviewer 3 Report
I have the following comments to improve the quality of the paper to be further considered for publication.
1) The paper is full of English grammar mistakes and bad sentence formations. I suggest the author use the English editing service or revise the manuscript properly with the help of a native English speaker.
2) In some places, the author has used the word fig. and in others the word figure. I suggest properly addressing the figure captions according to the journal format.
3) A performance table should be added to compare the proposed device performance with the existing devices in the literature.
4) There are a lot of abbreviations used in the paper, I suggest the author insert an Abbreviation section at the bottom of the paper.
5) Abstract section should be concise. It needs modification. There is no need to explain everything in this section. Provide only vital information and obtained results.
Author Response
Many thanks for your time with this manuscript; the authors value your comments very much. Based on the comments, the following revisions were effected in the manuscript.
- The manuscript has been thoroughly proofread to remove the language errors.
- Style of presentation has been made consistent.
- Regarding your suggestion to include a performance comparison between the previously published devices and the present ones, we believe that such comparison do not add much value to this manuscript. This is primarily because the variables involved in the cell fabrication, the efficiencies reported here is much lower than the state of the art devices. Nevertheless, a systematic comparison of the various devices with the control device clearly highlights the significance of the conclusions.
- Regarding the use of abbreviations, we tried to use them only essentially and all are defined at the first place.
- Abstract has been revised as recommended by popular style guides, which includes problem statement, methodology, results and their significance.
Reviewer 4 Report
In this paper, the authors introduced the photoelectric advantages of perovskite solar cells prepared by immobilizing plasmonic silver, copper and non plasmonic titanium particles on zinc oxide (ZnO) nanoflowers (NFS) scaffolds. This present study reveals promising avenues required in the field of plasmonic integration in photoanodes for the production of high efficiencies in PSCs. I believe that publication of the manuscript may be considered only after the following issues have been resolved.
1. What are the advantages of this job? It is suggested that the author give a table to compare the advantages of this work in detail.
2. Why is the RS difference of ZnO Cu so large? Is the gap between RCTs that big?
3. What is the corresponding relationship between the spectral absorption and photoelectric performance of the device? The author needs to point it out.
4. The introduction can be improved. The articles related to the some applications of Localized Surface Plasmon Resonance should be added such as Physical Chemistry Chemical Physics, 2022, 24, 4871 - 4880; Plasmonics 2015, 10, 1537–1543; RSC Adv., 2022, 12(13), 7821-7829; Plasmonics 2018, 13, 345–352; Appl. Phys. Express 2019, 12, 052015
5. The text information in Figure 7b is too small, and the author needs to make adjustments.
Author Response
Many thanks for your time with this manuscript. We have carefully considered your suggestions and revised the manuscript accordingly.
- The main advantages of this work is a systematic comparison of the the effect of plasmonic and non-plasmonic metal particles on the photocurrent density and open circuit voltage of perovskite solar cells. As stated in the first paragraph of the introduction, it read as "In this study, metal nanoparticles (Ag, Cu, Ti) /metal oxide nanostructures (ZnO) interfaces and examined the effect of plasmonic (Ag, Cu) and non-plasmonic (Ti) metal particles on the photocurrent density and open circuit voltage of PSCs".
- One possible reason for the large Rs in Cu ZnO could be loss of plasmonic properties with increase in particle size on account of its longer deposition time. However, a less pronounced variation was observed in Rct.
- As has been pointed out, an enhancement in the light harvesting efficiency due to absorption enhancement has contributed positively to the photocurrent density and open circuit voltage.
- Many thanks for pointing out these applications of plasmonic nanostructures, this has been effected in the last sentence of the second paragraph of the introduction, which read as "Besides, plasmonic particles are reported to be excellent light trapping devices when coupled with solar cells [68], light controlling in metal – dielectric – metal waveguides [69], design of tunable multi-band perfect metamaterial absorbers [70], highly sensitive refractive-index sensor based on strong magnetic resonance in metamaterials[71], etc."
- Thank you for suggesting to further discuss the IPCE data; the data has been further discussed, which can be read as "The corresponding IPCE curves in Fig. 7(b) exhibit the derived JSC data from J-V measurements and showcases the products of LHE, electron generation and collection efficiency of the solar device in the visible light range. IPCE for all the solar devices rise steadily from 300 nm and become stabilized in the 400 nm to 700 nm range and plummets at the 800 nm mark. ZnO-Ag5 boasts the highest IPCE with a reading of ~41% and trailed immediately by ZnO-Cu5 at ~39% while pristine ZnO NFs takes the middle ground with an IPCE of ~40 %. Strikingly, the ZnO-Ti group shows lower IPCE compared to the ZnO-Ag and ZnO-Cu pair by virtue of the non-plasmonic effects of Ti NPs. Owing to the increase in the LHE of the Ag analogues, corresponding enhancements in the PCE could be observed. The PSC fabricated using the ZnO-Ag 5 ETL has shown clear enhancements in the internal quantum efficiency at the violet and red ends of the spectrum. This enhancement in IPCE and internal quantum efficiency clearly increased the photocurrent density as shown in Fig. 7(a)."
Round 2
Reviewer 3 Report
The author has not properly answered my queries. I am not willing to accept the paper in its current form. I suggest the major revisions again.Kindly provide the abbreviation section, reduce the abstract section and make a comparison table.
Author Response
Dear Reviewer,
- The current abstract is written following published guidelines. There are many guides, the latest one (following which the current abstract is written) can be found in the Nature Index via https://www.natureindex.com/news-blog/artificial-intelligence-writing-tools-promise-faster-manuscripts-for-researchers
- As it is a research paper with limited size and limited number of abbreviations, it is a common practice to define at the first place. Indeed, the authors followed your suggestion to use them in moderation and define at the first place.
Reviewer 4 Report
Accept.
Author Response
Many thanks!
Round 3
Reviewer 3 Report
I am willing to accept the paper.